# A Painter in the Shadow: Unveiling Conservation, Materials and Techniques of the Unknown Luso-Flemish Master of Lourinhã

**Vanessa Antunes** [1,2,*], **Vitor Serrão** [1], **Sara Valadas** [3], **António Candeias** [3,4], **José Mirão** [3], **Ana Cardoso** [3], **Marta Manso** [2,5] **and Maria L. Carvalho** [2]

[1] ARTIS-Instituto História da Arte, Faculdade de Letras, Universidade de Lisboa (ARTIS-FLUL), Alameda da Universidade, 1600-214 Lisboa, Portugal; vit.ser@letras.ulisboa.pt

[2] LIBPhys-UNL, Laboratório de Instrumentação, Engenharia Biomédica e Física da Radiação, Departamento de Física, Faculdade de Ciências e Tecnologia, Universidade Nova de Lisboa, 2829-516 Caparica, Portugal; marta974@gmail.com (M.M.); luisa.carvalho@fct.unl.pt (M.L.C.)

[3] Laboratório HERCULES, Escola de Ciências e Tecnologia, Universidade de Évora, Largo Marquês de Marialva 8, 7000-676 Évora, Portugal; svaladas@uevora.pt (S.V.); candeias@uevora.pt (A.C.); jmirao@uevora.pt (J.M.); anamacardoso@yahoo.com (A.C.)

[4] Laboratório José de Figueiredo, Direcção-Geral do Património Cultural LJF-DGPC, Rua das Janelas Verdes 37, 1249-018 Lisboa, Portugal

[5] Faculdade de Belas-Artes, Universidade de Lisboa, Largo da Academia Nacional de Belas-Artes, 1249-058 Lisboa, Portugal

\* Correspondence: vanessahantunes@gmail.com

**Abstract:** The painting collection of Santa Casa da Misericórdia da Lourinhã is amongst Portugal's most notable and scarcely best-known cultural heritage. The artistic interest of this pictorial group, besides the advanced state of degradation of a number of the paintings, together with the ruined circumstances of the building accommodating the collection, today in reconstruction, were the key reasons for this study. Thermo-hygrometric measurements were carried out. A multianalytical methodology incorporating micro-X-ray diffraction (μ-XRD), energy-dispersive X-ray fluorescence spectroscopy (EDXRF), scanning electron microscopy–energy dispersive spectroscopy (SEM–EDS), micro-Raman spectroscopy (μ-Raman), micro-Fourier transform infrared spectroscopy (μ-FTIR) has been followed for the study. These analyses were complemented by infrared photography (IRP) and reflectography (IRR), allowing the study of the underdrawing technique. The results of this study were compared with previous ones of the painter's workshop and important distinctions and similarities were found within the materials and techniques used. This analysis methodology on materials contributes to safeguarding and the ensuing community awareness of this cultural heritage in danger.

**Keywords:** easel painting; conservation of Cultural Heritage; Master of Lourinhã; ground layer; multianalytical methodology

---

## 1. Introduction

The painting collection of the Santa Casa da Misericórdia da Lourinhã (SCM Lourinhã) is one of the most remarkable and little-known elements of cultural heritage of Portugal. The artistic interest of this pictorial group, together with the advanced state of degradation of some paintings and the ruined conditions of the building that houses the collection, currently under reconstruction, were the main reasons for this study. The main objective of this work is to reveal knowledge about this important

part of national heritage, their authors and their workshops, by studying the artistic and material characterization and the state of conservation of this collection.

Such a methodology of research on materials and its conservation contributes to the safeguarding and consequent awareness of the endangered cultural heritage of the community.

The research on the two Renaissance paintings representing "São João Baptista no Deserto" (P1) and "São João Evangelista em Patmos" (P2) (Figure 1), originally from the extinct Hieronymite Monastery in Berlengas Island (Peniche, Portugal), is enough to credit the artistic qualities of the collection and to justify the urgency of an integrated project study, conservation and musealization. The paintings were transferred from the Convent in Berlengas Island, still in the 16th century, to the monastery of Vale Benfeito (Óbidos), when this monastery was abandoned due to its inhospitality. The painting collection was later deposited in Santa Casa da Misericórdia da Lourinhã during the exclaustration of Vale Benfeito monastery.

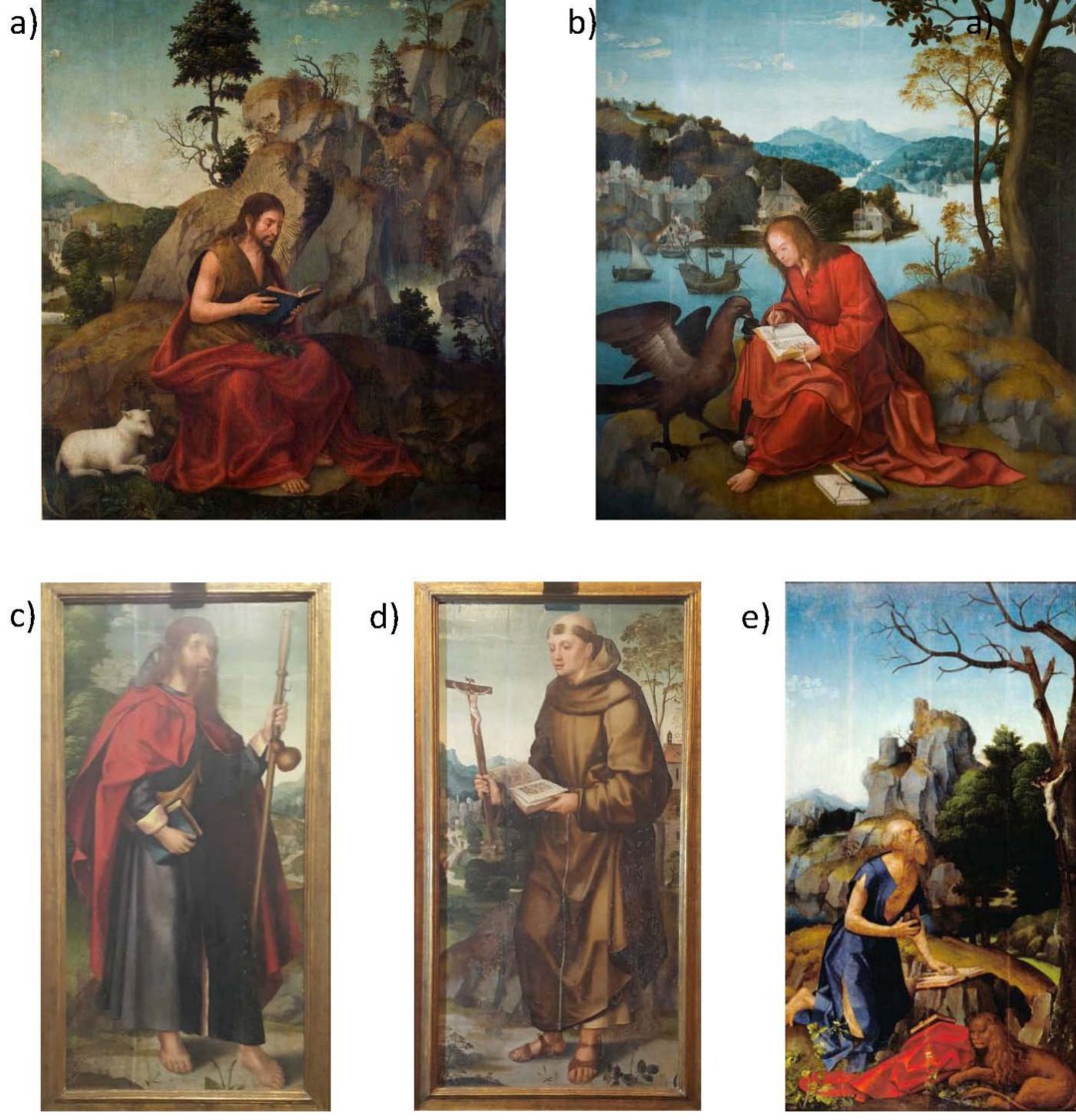

**Figure 1.** (**a**) "São João Baptista no Deserto" (P1); (**b**) "São João Evangelista em Patmos" (P2); (**c**) "S. Tiago" (P3); (**d**) "Sto.António" (P4); (**e**) "S.Jerónimo"(P5).

These masterpieces are the best that Renaissance painting produced in the governance of King D. Manuel I in the Luso-Flemish period. The paintings were commissioned c. 1515 by Queen D. Maria, second wife of King D. Manuel I. The underdrawing, the fineness of the landscapes and the transparency of chromatic material, reveal the talent of a great artist following the Bruges and Antuerpian models. Brought to light by Luís Reis-Santos in 1937 while visiting the Santa Casa da Misericórdia da Lourinhã, these panels are due to an artist who has taken since then the name of Master of Lourinhã [1], an epithet that brings together a series of 30 royal paintings in the same Luso-Flemish manner [2–4].

In this research, the material and technical results on the two Lourinhã paintings are compared to other two unstudied paintings from the former Cabo Espichel chapel, "S. Tiago" (P3) and "Sto. António" (P4) assigned to the same master, and to other paintings such as Penha Longa (Sintra), "S.Jerónimo"(P5) (Figure 1), the Funchal Cathedral (Madeira island) altarpiece collaborative work, and also to the work of Frei Carlos, to whom the Master of Lourinhã paintings were formerly assigned.

Considering that this study concerns cultural heritage artifacts, the analytical techniques employed were useful to derive important non-destructive and non-invasive measurement information on elementary composition of the materials used by color, specifically addressing X-ray fluorescence spectroscopy (XRF) analysis. These results were crucial to define the sampling areas, avoiding overpaintings. Elementary analysis was complemented by sampling the main colors pallete for each painting, and performing scanning electron microscopy–energy dispersive spectroscopy (SEM–EDS) in cross-sections, allowing us to understand elementary distribution through layers. This technique was also useful to recognize the distribution of the grains, their morphology and the granulometry of the ground layer through backscattered electron images (BSE). Ground layer study was essential to determine the material and technical tendencies followed by the artist [2]. Compound analysis on this layer was performed by μ-Raman technique, complemented by micro-X-ray diffraction (μ-XRD) whenever needed, to clarify the relative proportion of gypsum/anhydrite facing a calcium carbonate upper layer (P5).In order to understand stratigraphic layering by color, optical microscopy (OM) was used, and micro-images were taken to compare layering technique by color and by painting. Afterwards, the painting components for each layer were analyzed by μ-Raman technique, in order to recognize the compounds and their state of conservation. These analyses were confirmed by μ-FTIR when hardly identified by the μ-Raman technique, as with the carbon black or the green pigments, a difficulty already noticed by other studies on historical samples [3,4].

Area infrared photography (IRP) and infrared reflectography (IRR) exams were useful to observe carbon density in drawing by technique and painting, and comparing between paintings, since the same equipment and performance levels were used. These area exams also allowed us to distinguish different underdrawing technique for each painting and permitted the comparison between the paintings.

The outcomes achieved by the combination of different analytical procedures unveil the materials and the techniques of the unknown Luso-Flemish painter Master of Lourinhã and bring information on materials conservation, an essential issue concerning cultural heritage preservation.

## 2. Materials and Methods

### 2.1. Sample Collection and Preparation

Visual examination, infrared Reflectography and photography on the paintings with the goal of searching the state of conservation of the paintings were performed. The elementary micro-XRD technique enabled the identification of sampling areas by painting, avoiding overpaintings. One sample of 200–300 μm by color was taken in similar areas for each painting. This allowed us to identify the components by color and by layer and its state of conservation. It also allowed comparing color palettes between paintings. Samples were partly mounted as cross-sections in epoxy chemical compound resin and polished with carbide. One part of each sample was kept to be analyzed by micro-Raman (μ-Raman).

*2.2. Description of the Analytical Techniques*

Samples were studied by optical microscopy (OM) with a Leitz Wetzlar optical dark field and bright field magnifier microscope, including a photographic camera Leica DC 500.

Macro-photographs were captured with a mobile microscope 3" LCD 8.5 Mega Pixels 20-500x, Digital LCD with VGA, Micro SD card storage and a MicroCapture Pro software.

Examination by IR photography was obtained with a digital camera SONY DSC-F828, 7 Mega Pixels.

Infrared reflectography was performed with a high-resolution infrared reflectography camera (Osiris) with an InGaAs detector allowing a wavelength response from 900 to 1700 nm and equipped with a 16 x 16 tile system that allows a picture size of 4096 x 4096 pixels. The camera comes with a long pass filter Schott RG850, permitting it to transmit the infrared wavelength and block the unwanted shorter wavelength till 850 nm. The reflectograms were performed for $45 \times 45$ cm$^2$ painting´s area.

Portable energy-dispersive X-ray fluorescence spectroscopy (EDXRF) used an Amptek Mini-X Rh X-ray generator, 50 keV, 200 mA, 2.25 W, and an Amptek XR-100SDD silicon drift detector with a 25 mm$^2$ detection area and 500 mm thickness and a 12.5 µm Be window. A 1 mm collimator was used to perform the analyses. The energy resolution was 140 keV at 5.9 keV. The angle between the incident and therefore the emitted beam was 90°. This geometry permits a high background reduction thanks to Compton scattering. The X-ray generator was operated at 30 kV and 15 µA throughout 120 s. The analysis was carried out in air atmosphere. Spectra were performed using DppPMCA package and its deconvolution and analysis were performed using WinAXIL software package by Canberra. Scanning microscopy with SEM–EDS was performed by a Hitachi S-3700N scanning microscope with a coupled Bruker XFlash 5010 SDD energy-dispersive detector in operation at 20 potential units was used to perform scanning microscopy imaging (backscattering mode) and elementary composition of the analyzed samples cross-sections was obtained by SEM–EDS, in operation at 20 kV. Samples were analyzed in variable pressure while not carbon coating to be any analyzed by different analytical ways.

µ-XRD was performed using a Bruker general area detection diffraction system (GADDS) microdiffractometer (Bruker AXS, D8 Discover) which was accustomed to performing micro-X-ray diffraction. This microdiffractometer is provided with a two-dimensional HiStar gas stuffed area detector, a Goebel mirror, a laser-video sample alignment system and a motorized XYZ stage. Diffraction data were registered using Cu K$\alpha$ radiation, a tube running at 40 keV, 40 mA, with the incident beam collimated to 1 mm diameter. XRD patterns were measured in the 2θ range between 8° and 70°. A step size of 0.02° was performed with a recording time of 1800 s for every step. International Centre for Diffraction Data powder diffraction files (ICDD PDF) were used for the identification of crystalline phases using the Bruker EVA package. Samples were analyzed unmounted. Micro-Raman spectroscopy (µ-Raman) analyses were undertaken by employing a Horiba–Jobin Yvon XploRA confocal spectroscopy equipment, using a 785 nm excitation wavelength, with incident power of 0.2 mW. Employing a 100 × magnification objective with a pinhole of 300 µm and an entrance slit of 100 µm, the scattered light-weight collected by the target was spread onto the air-cooled CCD array of an Andor iDus detector by a 1200 lines/mm grating. Raman spectroscopy was performed in a range of 100–3000 cm$^{-1}$ with laser spot diameter 1 µm. Spectra deconvolution was performed using LabSpec (V5.78). The identification of pigments was created with Spectral ID$^{TM}$. Micro-Fourier transform infrared (µ-FTIR) spectroscopy was performed in a microscope Hyperion 3000 controlled by software OPUS 7.2 from Bruker and a Mercury Cadmium Telluride detector are coupled to a Bruker spectrometer Tensor, 27 model. A medium infrared region (MIR), in transmission mode, with a 15x objective and a diamond compression microcell EX'Press 1.6 mm, STJ-0169 were used. For each spectrum, a spectral resolution of 4 cm$^{-1}$ was performed. A working range of 4000–600 cm$^{-1}$ and 64 scans were recorded.

Thermo-hygrometric measurements were performed with a psychrometer of the Brannan brand (England) and confirmed with manual and digital thermometers. An evaluation was performed through a psychrometric scale relating room temperature and humidity.

## 3. Results and Discussion

### 3.1. State of Conservation of the Paintings

A micro-FTIR spectrum from the green shadow of the grass, in P2, displays azurite, kaolinite, oil and metallic carboxylates. Results from the light green of the grass, in P4, display the same results, apart from kaolinite (Figure 2). The presence of metallic carboxylates, confirmed also by μ-Raman, such as plumbonacrite, indicate the need to stabilize chemically these soaps since they are responsible for a continuum of chemical degradation phenomena [5]. These chemical reactions also contribute to the physical degradation of the paintings, since the increase of volume of lead carboxylates causes instability in the ground and painting layers, being probably one of the causes contributing to the detachment of these layers from the support.

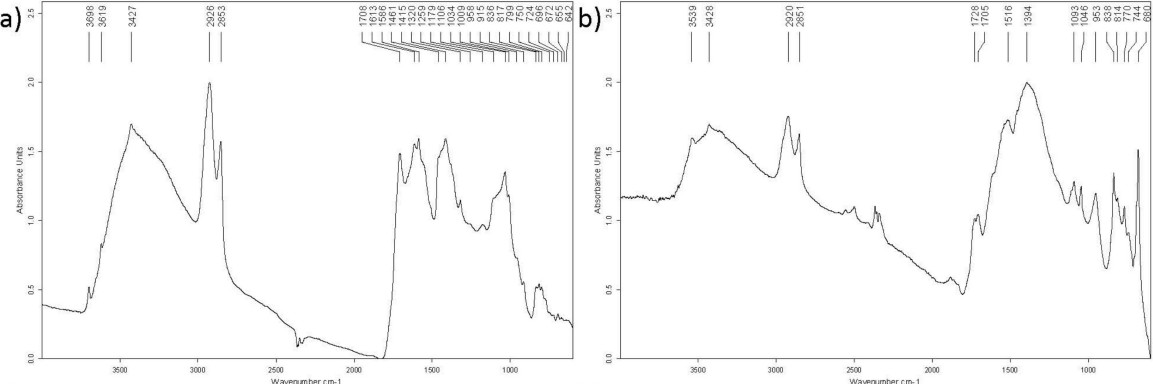

**Figure 2.** (**a**) Micro-Fourier transform infrared (μ-FTIR) spectrum from the green shadow of the grass, in P2, displaying azurite, kaolinite, oil and metallic carboxylates; (**b**) μ-FTIR spectrum from the light green of the grass, in P4, displaying azurite, hydrocerussite, oil and lead carboxylates.

These results prove the evidence of the poor state of conservation of the studied groups, with no exception. Big areas of lacunae were detected in many parts of the paintings, some of them already restored (P1–P2, P5). These lacunae are more evident in the lower part of each painting as confirmed by IRR and IRP results. This fact suggests that a higher amount of humidity was accumulated in this area. Ascensional absorption of the wood canals in the lower part of each painting was the main cause for this degradation, probably being in contact with floor humidity. Also, the high humidity in the air was verified. Relative humidity was controlled in P1 and P2 in summer and winter. In other paintings (P3–P5) climate verification was not possible for this study, since it takes at least one year to analyze climate changes according to seasoning. P3 and P4 were brought from a church to the museum climate during the last year and P5 remained for several years in a museum climate. Results on P1–P2 climate verification showed that the change in the relative humidity environment varies between 63% and 70% in the dry season (summer). This means a variation of 7% in relative humidity between morning and afternoon, and the relative humidity should not fluctuate more than 10% in 24 hours. It has to be considered that this maximum value of 70% of relative humidity is the value from which it is agreed that the various objects of organic structure, especially painting, begin to change their dimensions, lose their original rigidity, becoming plastic and more vulnerable to the formation of fungus [6,7]. However, in the wet season (winter) an approximate relative humidity of 78%–83% was detected.

The 6%–7% variation in relative humidity between the morning and afternoon measured periods, both in the cold and the hot season, leads to the assumption that the room oscillations should not exceed 10% in the 24 h. This aspect has a high importance to the stability of the paintings, since, according to recent studies, the possibility of wood degradation and polychrome with oscillations greater than ± 10% of the relative air humidity and temperature oscillations, greater than ± 10 °C, can cause physical deterioration in the chromatic layers and varnishes [6].

The temperature of the room in the summer season varies, between 24 °C and 26 °C. Taking into account the studies on these values, the temperature should always be below 30 °C, since for higher values the probability of deterioration of the materials constituting the works (glues and adhesives) increases. Therefore, in the context of the exhibition, it is recommended by some authors that a temperature below 30 °C is assumed, considering that is a compromise between the stability of the works and the thermal comfort of visitors [6]. In the winter season temperatures between 10 °C and 13 °C were detected. Several risk factors were detected in the room exhibiting P1 and P2. Natural factors (Meteorological, Hydrological, Seismic and Building Ruin Conditions) and Disaster derived from human activity (River protection failure and riverbank drainage, Building protection failure -easy to theft and vandalism, failure of poor electrification -over the wooden ceiling, failure of roof and ceiling maintenance (Table 1).

**Table 1.** Analysis summary of risk factors: natural and derived of human activity in the room (adapted from [7]).

| Risk | Natural Disaster | Disaster Derived from Human Activity | Indirect/Secondary Disaster |
|---|---|---|---|
| **Meteorological** | Rays<br>Intense precipitation<br>-Strong winds<br>Fire from thunderstorm lightning | | Fluvial flood<br>Fire |
| **Hydrological (caused by heavy rainfall)** | Flood caused by precipitation (insufficient drainage and infiltration) | River protection failure and riverbank drainage | Biological contamination due to high humidity |
| **Seismic** | Tectonic plate movement | | Drop of the building<br>Fire<br>Inundation |
| **Building Ruin Conditions** | Fall of building through meteorological, hydrological or seismic hazards | Building protection failure (easy to theft and vandalism)<br>Failure of poor electrification (over wooden ceiling)<br>Failure of roof and ceiling maintenance | Full roof<br>Fire<br>Inundation<br>Destruction of the collection |

### 3.2. Technical and Material Characterization

#### 3.2.1. "São João Baptista no Deserto" (P1) and "São João Evangelista em Patmos" (P2)

The ground of both panels is similar, having more anhydrite than gypsum (*gesso grosso*) spread in a double layer of thicker grains in the base and thinner grains in the top. This sort of double ground layer is frequent in Lisbon painting workshops and particularly tending to the past examined work of Jorge Afonso's workshop [8]. The use of minium in the ground layer was probably a way to give the ground layers a less white and warmer tone to paint since gypsum has a significant refractive index when compared to chalk [9].

Moreover, in this pair of paintings, the calcium sulfate-based ground contains minium grains (Tables 2 and 3).

Underdrawing is characterized by the first stage of geometrical drawing by incision and dry charcoal and a second phase of fluent brush lines of additive parallel traces in brownish-black ink to provide the forms and shadow areas. It is also possible to see by IRR some corrections to the forms, such as in the position of the hands of the saints.

Underdrawing has a first phase of a geometrical point made by incision and by dry charcoal, as is evident in the face of "São João Baptista no Deserto" (P1) where a geometrical trace marks the nose position (Figure 3).

Low contrast in IRR results, characterizing the underdrawing of these paintings is probably due to a low content carbon-ink, e.g an iron-based ink with a small quantity of carbon, as identified in other Portuguese-Flemish paintings [10–12].

Master of Lourinhã creativity is visible in the sketchy underdrawing, characterized by a spontaneous freehand drawing. Occasionally with changes of composition and given up drawings, the artist had the draftsman skills to block the final composition, suggesting that the sketching on a white ground layer was well known to this Luso-Flemish artist.

**Table 2.** Main pigments by the color of the paintings P1-P4 assigned to Master of Lourinhã with results by X-ray fluorescence spectroscopy (XRF) and μ-Raman.

| Colors. | Pigments by Colors | Micro-XRF Key-Elements (P1-P4) | Micro-Raman Key-Compounds (P1–P4) |
|---|---|---|---|
| Yellow | Lead-tin yellow | Pb, Sn | yellow ochre, cerusite, cinnabar, carbon black, lead-tin yellow type I |
|  | Ocher | Fe |  |
|  | Vermilion | Hg |  |
| Blue | Azurite | Cu | azurite, cerusite |
| White | Lead white | Pb | lead white, carbon black |
| Brown | Ocher | Fe | lead white, carbon black, cerusite, minium, cinnabar |
|  | Vermilion | Hg |  |
| Black | Vegetable charcoal | - | carbon black |
|  | Animal Charcoal |  |  |
| Green | Green | Cu | cerussite, gypsum, azurite, cinnabar, led-tin yellow type I, carbon black |
|  | Green Dye |  |  |
|  | Lead-tin yellow | Pb, Sn |  |
|  | Azurite | Cu |  |
|  | Malachite | Cu |  |
| Red | Red Ocher | Fe | cinnabar, carbon black, hematite |
|  | Vermilion | Hg |  |
|  | Minium | Pb |  |
| Flesh tones | Ocher | Fe | cinnabar, plumbonacrite, goethite, carbon black, hematite |
|  | Vermilion | Hg |  |
|  | Lead white | Pb |  |
|  | Lacquer | - |  |
| Pre-gilding layer | Ocher | Fe | minium, lead white, quartz |
| Ground layers | Gypsum | Ca | minium, anhydrite, gypsum |
|  | Anhydrite | Ca |  |
|  | Minium | Ca |  |
|  | Calcite (P5) |  |  |

A probable iron-based ink underdrawing in P1 and P2 connects Master of Lourinhã materials to Frei Carlos paintings. A heterogenic underdrawing characterizes Frei Carlos works [13]. A scarce drawing in certain works is contrastive with other paintings having an abundant drawing [10,11]. IRR shows that single iron-based ink or mixed with carbon was used in some of the painter's works [14].

**Table 3.** Micro-FTIR and μ-Raman wavenumbers of compounds found in P1–P5 and database comparing source.

| Compounds | Micro-Raman Characteristic Peaks (cm$^{-1}$) | Micro-FTIR Characteristic Bands (cm$^{-1}$) | Database Comparing Source |
|---|---|---|---|
| Azurite | 246, 281, 327, 397, 471, 541, 768, 831, 941,1097 | 3425, 1464, 1412, 1092, 956, 837, 817, 769, 747, 694 | Infrared and Raman Users Group Spectral Database (IRUG) |
| Carbon black | 1322, 1580 | | Artists' Pigments |
| Minium | 220, 314, 391, 551,465,1053 | | Artists' Pigments |
| Cerusite | 259, 670, 827, 1057 | | IRUG |
| Hydrocerussite | | 3534, 1045 | IRUG |
| Cinnabar | 253, 282, 343 | | RRUFF Project (RRUFF) |
| Goethite | 256, 302, 388, 470,550 | | RRUFF |
| Gypsum | 373, 416, 956, 1019 | | IRUG |
| Hematite | 224, 253, 294, 412, 495, 612 | | Artists' Pigments |
| Kaolinite | | 3695, 3685, 3670, 3652, 3620, 1114, 1033, 1011, 914, 796, 754, 696, 657 | IRUG |
| Lead carboxylates | | 1541 | Artists' Pigments |
| Lead-tin yellow type I | 123, 192, 276, 367, 451, 536 | | Artists' Pigments |
| Oil | | 2926, 2855, 1740, 1462, 1238, 1159, 1099, 722 | Artists' Pigments |
| Plumbonacrite | 319,419,1053 | | RRUFF |
| Quartz | 212, 353, 395,465 | | IRUG |
| Yellow ochre | 258, 303, 401, 477, 555 | | IRUG |

A recipe from an Hebraicized Portuguese manuscript, "As cores", probably from the 15th century [15,16], marks the manufacture of a recipe of ferrous-base ink [17]. Also, studies on coeval Flemish illuminated books reflect the use of this iron-based ink in the making of the drawing and on the shadow tones [18]. A Portuguese treatise on painting from the beginning of the 17th century still delivers a recipe of iron-based ink, evidencing the big utility of this material for draftsmen, illuminators, and painters [19].

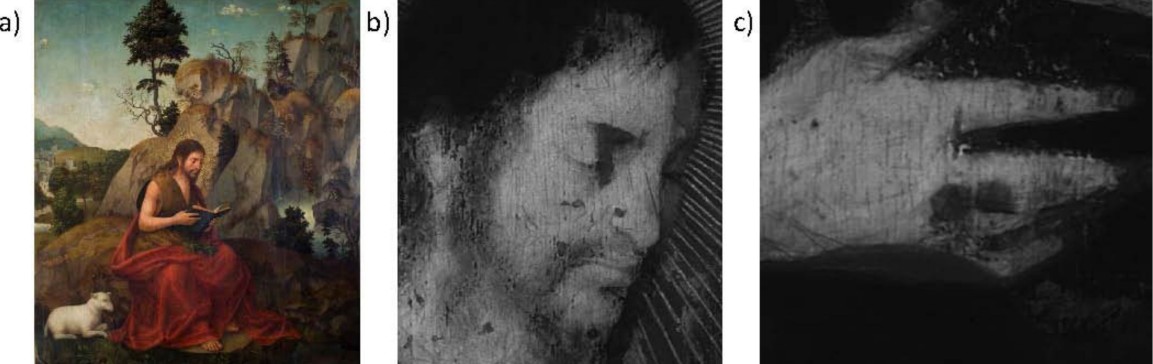

**Figure 3.** (**a**) "São João Baptista no Deserto" (P1); (**b**) underdrawing in geometrical trace marking the nose position u; (**c**) corrections to the form of the hand of the saint (infrared reflectography (IRR)).

The collaboration of the painter with other luso-flemish masters has been proposed in previous studies. Considering the Flemish origin, working in Portugal in coeval years, using the same materials and techniques, it is possible to infer collaboration between both painters. However, when comparing both draftsmens' work, significant differences are highlighted, such as the hands of the figures, drawing the fingers in a pointy finishing, in the case of Master of Lourinhã.

Priming is particularly tended to be thicker in certain areas such as the blue color of the sky, the landscapes, or the tunics and mantles. Restricted areas were left for the painting of the figures, such as in the case of the griffon wings, changed in the final painting (Figure 4).

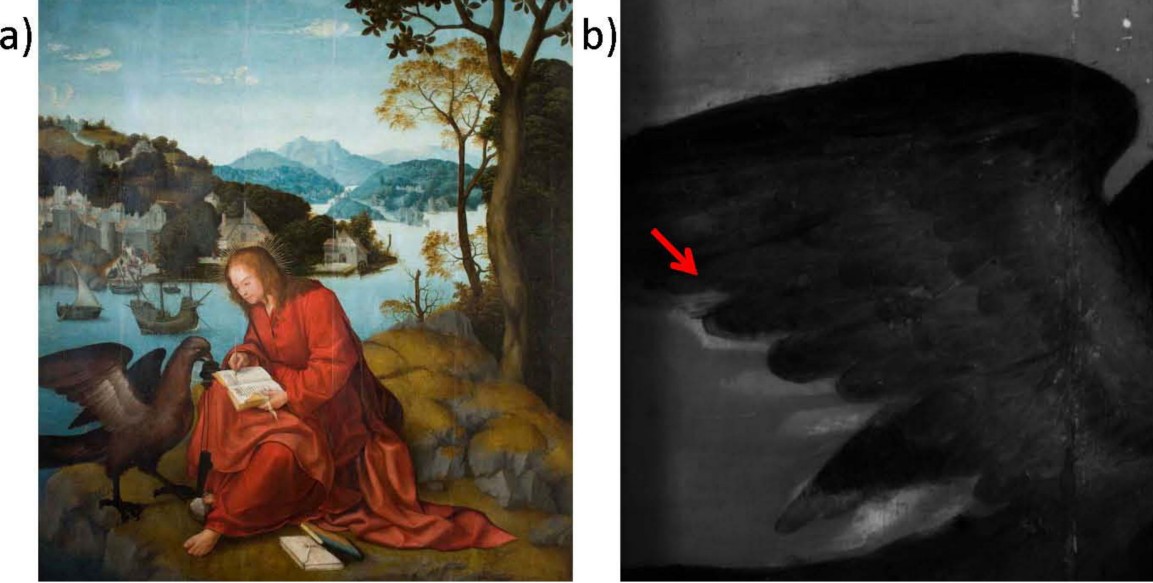

**Figure 4.** (**a**) "São João Evangelista em Patmos" (P2); (**b**) griffon wings, changed in the final painting (IRR).

Tables 2 and 3 summarize the pigments found by color in these studied paintings: yellow (lead-tin yellow, ocher, vermilion; blue (azurite); white (lead white); brown (ocher, vermilion, umber); black (vegetable charcoal, animal charcoal); green (green dye, lead-tin yellow, azurite, malachite); red (red ocher, vermilion, minium); flesh tones (ocher, vermilion, lead white, lake) [20–22]. Few gilded areas are found in the paintings, restricted to the saints halos. The pre-gilding pigment layer is probably made with mordant. Taking into account the characteristics of the pictorial stratum, in oil composition, this binder was also used as mordant containing pigments in order to give volumetry to the layer (ocher) (Figure 5). Changes in Raman position were verified in carbon black, cerusite and hematite. Raman spectra results evidence that carbon black is changing from 1322 to 1370 cm$^{-1}$ peak position. This indicates that these carbon black materials have a degree of disorder in the typical range of graphite rod and graphite powder [23]. Changes in cerusite main peak are also from 1055 to 1057 cm$^{-1}$ peak position, being their main symmetric C–O stretches very close to the 1054 cm$^{-1}$ band, which may show deformation of the carbonate tetrahedra [24]. Changes in hematite peaks may be due not only to each specific particle size, but also to the growth kinetics, both determining the structure of hematite nanoparticles [25].

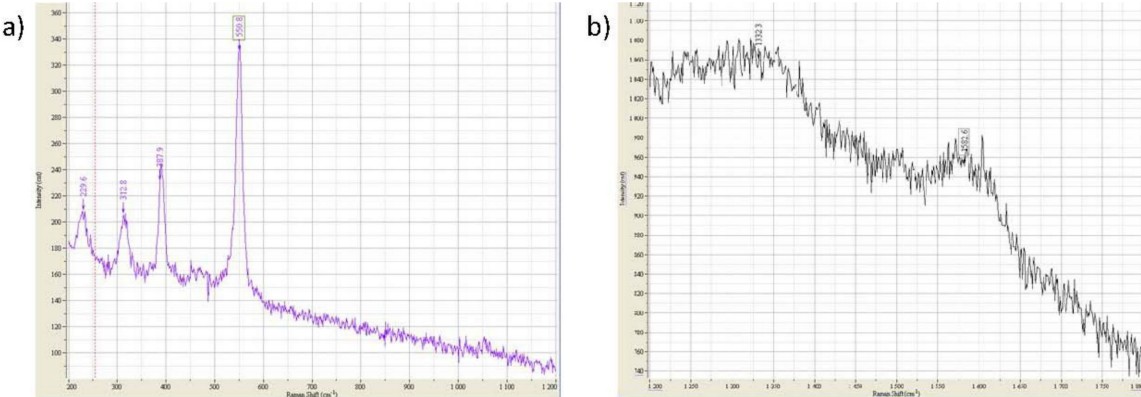

**Figure 5.** (**a**) Micro-Raman spectrum displaying in P1, in the region of the pre-gilding layer, under the golden halo, the presence of minium (226, 315, 390, 551cm $^{-1}$); (**b**) Micro-Raman spectrum displaying in P2, in the region of the black book, the presence of carbon black (1333, 1583 cm $^{-1}$).

The technique of shadows / shinning areas was made including particular colors of brown/ochre, used not only to the contour of the main figures in the flesh tone areas, but also to paint minuteness figures, as illuminators painters did, like in the manuscript Illumination with Scenes from the Life of Saint John the Baptist, made in Bruges, Netherlands, circa 1515, assigned to Master of James IV of Scotland, probably Gerard Horenbout (South Netherlandish, active ca. 1485–1530) [18,26] (Figure 6).

Medium tones were achieved utilizing diverse concentrations of the same materials for the color-making. Lighter tones were achieved in certain areas by hatching parallel white brushstrokes, as occurs in illuminated books [18].

Master of Lourinhã studied works (P1–P5) deliver the characterization of the artist's palette by the following core-pigments: for the red color the vermilion matrix; for the brown color ochre and vermilion; for the blue color, an azurite matrix; golden halos made in ochre and gold; for the green color, copper green (malachite, verdigris or copper acetate) and lead white. Flesh tones, composed by lead white, vermilion, ochre, and red lacquer; when compared between the studied groups P1–P5 have similar material admixture. Tables 2 and 3 show the compilation of the artist's palette characterization, concurring to the most pigments found by color.

It is noteworthy that the use of a more grayish layer under the final flesh tone is common to the studied panel paintings and manuscript illumination, rendering the shadows and turning the flesh tone more realistic. The last flesh tone layers were then applied in thin, translucent and diluted layers to profit the vibration of the previous color. Many artists at the time painted in both supports, using similar techniques. It is the case of the celebrated Flemish painters Gerard David, or Gerard Horenbout [18], both having masterpieces in Portugal. This last one, also known as Master of James IV of Scotland, can be connected to Master of Lourinhã paintings by its techniques and close aesthetic values. The intense luminosity of the paintings under study, the deep transparencies, and water reflection effects, or the delicate minuteness in the quick making of small figures, in secondary scenes, spread through architectonic and natural Nordic landscapes evidencing geometrical rocky cliffs, approaches both masters work. Also, the fact that both masters used the delineation with the brownish color of the flesh tones as a modeling technique, similar to underdrawing tone resembling the illuminated manuscripts technique used by Horenbout [6], one of the draftsmen of the breviary of queen Leonor of Portugal, one of the wealthiest queens of her time.

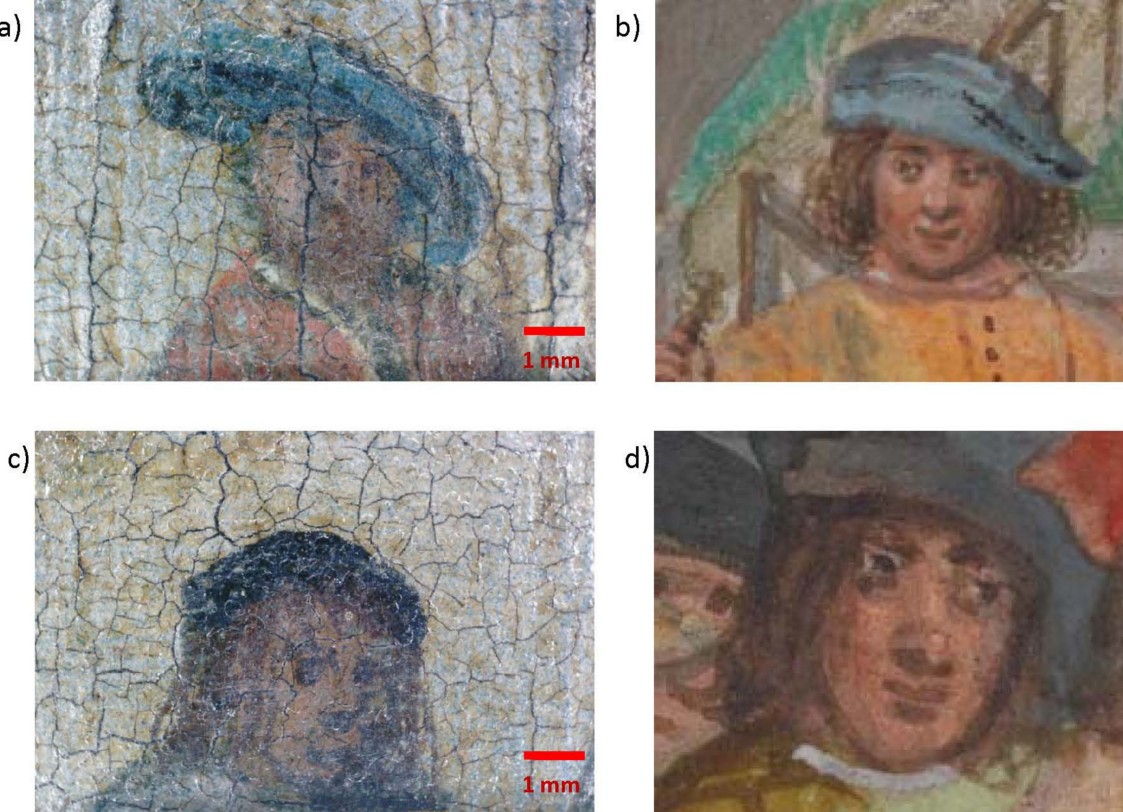

**Figure 6.** (**a**,**c**) Minuteness figures in brown contour in the painting "São João Evangelista em Patmos" (P2); (**b**,**d**) Minuteness figures in brown contour in the manuscript Illumination with Scenes from the Life of Saint John the Baptist, Made in Bruges, Netherlands, ca. 1515, assigned to Master of James IV of Scotland, probably Gerard Horenbout, Metropolitan Museum of Art.

Although, contrary to Frei Carlos's Catholic profession when arriving in Portugal, a predominantly Catholic country, such as Flanders, Gerard Horenbout was a Lutheran, moving to England circa 1522-25, to work for a wealthy clientele and after to the royal court [27]. Coincidently, the Master of Lourinhã corpus of paintings is placed in the first half of the 16th century.

Pigments by color displayed in Table 2 are comparable to the ones found in collaborative work between Master of Lourinhã, the Lisbon painting workshop leader, Jorge Afonso, and the Flemish painter Francisco Henriques, in the case of the Funchal Cathedral (Madeira Island, Portugal) altarpiece. Similar pigments were found in this case [28]. Also, Frei Carlos paintings display a similar palette [14], apart from the intentional use of brochantite green [17,23], a pigment not found in Master of Lourinhã studied paintings P1–P4 (Figure 2).

### 3.2.2. "S. Tiago" (P3) and "Sto.António" (P4)

Ground layers of both paintings are composed of calcium sulphate, having more anhydrite than gypsum (*gesso grosso*).

The infrared photography of this pair of paintings shows that a carbon-based ink was used to perform the underdrawing. The first planned eyes direction of the saints, looking down, was changed in the final painting of the two panels, looking ahead, probably to coincide with the previous models, as it is possible to deduce from Figures 7 and 8. Models of illuminated books by Horenbout workshop seem to have been used, as it is the case of St James in the Book of Hours of James IV and Margaret Tudor, 1503, or of the illuminated image of Saint Anthony of Padua, belonging to the group manuscript of the Spinola Hours, made ca. 1510–1520.

Lighter tones were achieved in certain areas by hatching parallel white brushstrokes, to finalize the highlighting of the figures, as occurs in illuminated books [18] (Figure 9).

Concerning the main pigments and based on the results obtained by EDXRF and Raman spectroscopy, lead white is the base of the white color and priming. It is also used in admixture to other pigments to achieve the lighter hues. Malachite is the matrix for green color; vermilion is the basis for the red color; ochre and lead-tin yellow was matrix for yellow colors, this last being also used in the changing hues of "São Tiago" (P3) tunic; ochre and vermilion were used for the brown color and azurite as the matrix for blue color. The delineation of the flesh tones in brown color is visible, and the small miniaturized figures in the landscapes are frequent in these two paintings, using, moreover similar characters to P1 and P2.

Under the flesh tones was applied a brownish-grayish modelling layer to contrast in the final composition. This technique is also visible in the illuminated minuteness figures in the Scenes from the Life of Saint John the Baptist, made in Bruges, Netherlands, ca. 1515 (Figure 10).

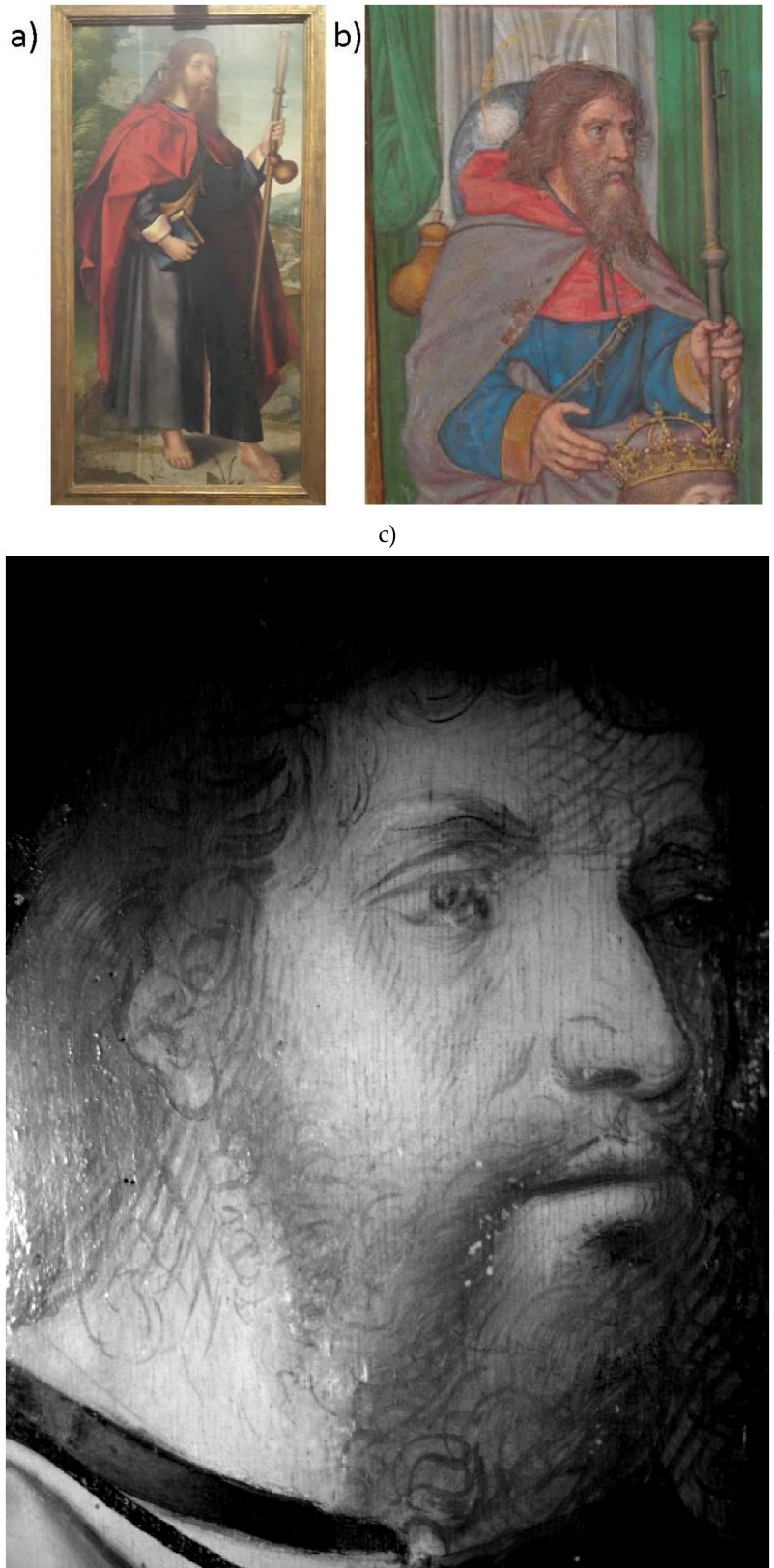

**Figure 7.** (**a**) "S. Tiago" (P3); (**b**) detail from the Book of Hours of James IV and Margaret Table 1503. Österreichische Nationalbibliothek, Vienna; (**c**) IRR detail, the first planned eye direction of the saints was changed in the final (P3).

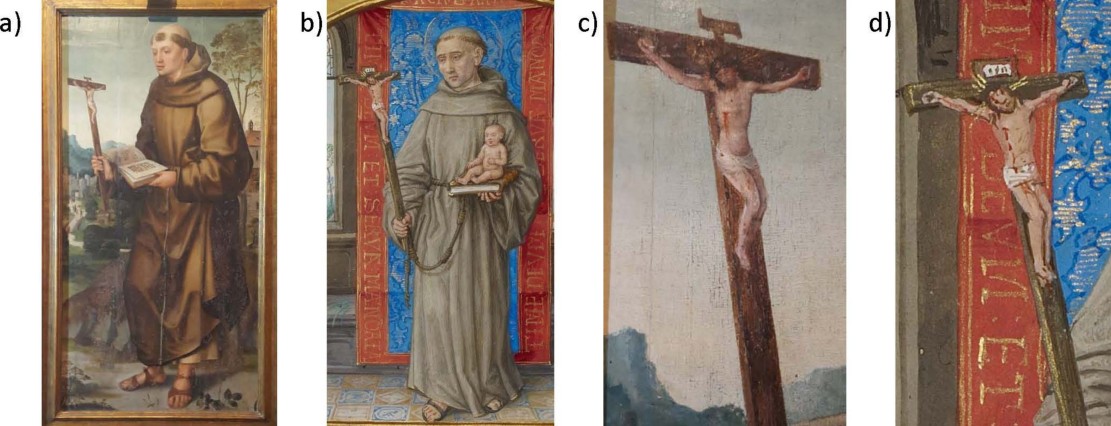

**Figure 8.** (**a**,**c**) "Sto. António" (P4); (**b**,**d**) *Saint Anthony of Padua*, Spinola Hours, ca. 1510–1520, Master of James IV of Scotland, Getty Museum.

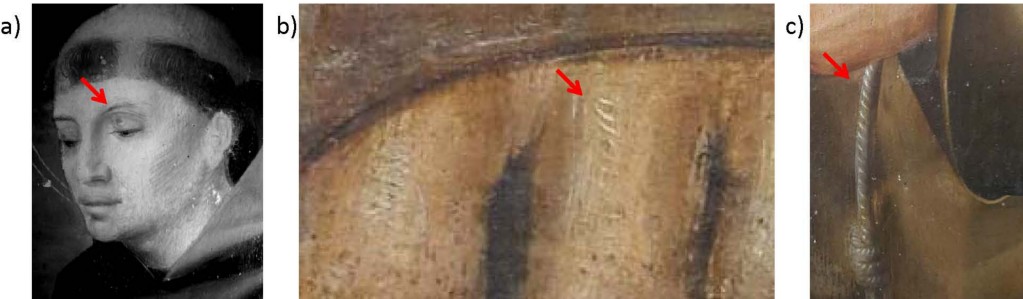

**Figure 9.** (**a**) Detail first planned eyes direction of the saint changed in the final (P4); (**b**,**c**) detail of hatching parallel white brushstrokes to achieve lighter areas.

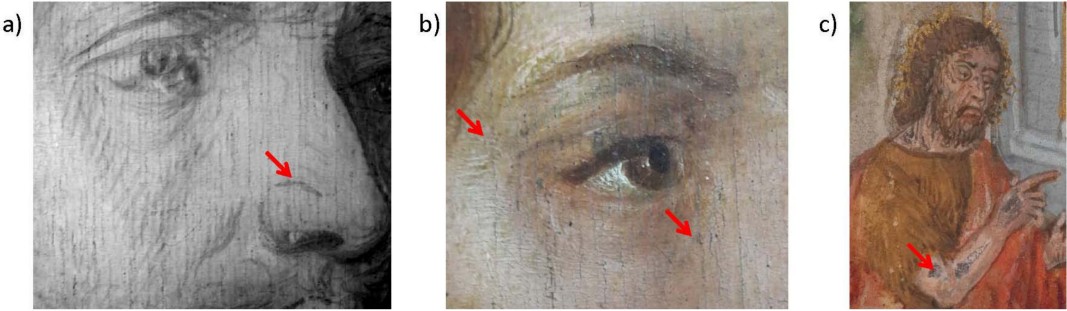

**Figure 10.** (**a**) IRR detail contour drawing in the nose and first planned eye direction of the saint, changed in the final (P3); (**b**) brownish-grayish layer under the flesh tone of the figure (P3); (**c**) brownish-grayish layer under the flesh tone of the figure in the manuscript Illumination with Scenes from the Life of Saint John the Baptist, Made in Bruges, Netherlands, ca. 1515, assigned to Master of James IV of Scotland, probably Gerard Horenbout, Metropolitan Museum of Art.

### 3.2.3. "S.Jerónimo"(P5)

The painting's ground is made of calcium sulfate, is a *gesso grosso* ground layer (with more anhydrite than gypsum) (Figure 11).

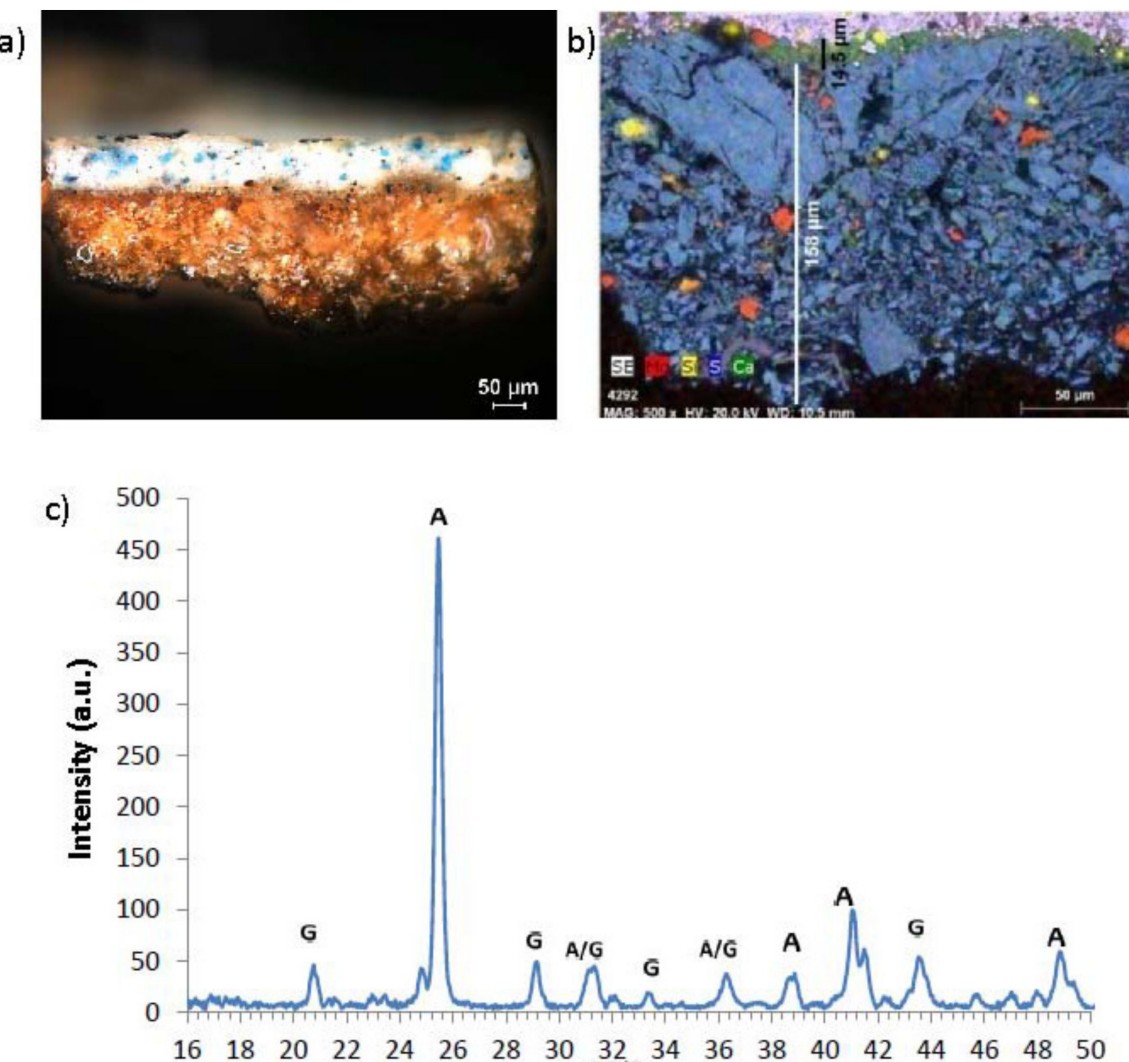

**Figure 11.** (**a**) Optical microsocpy (OM) of sample of the rock grey in P5; (**b**) elementary analysis by scanning electron microscopy–energy dispersive spectroscopy (SEM–EDS) of the same sample showing the thickness of the calcium sulfate ground layer, in blue, as well as the thin layer of calcium carbonate that overlaps it, in green; (**c**) diffractogram of the chemical analysis of the lower layer of sample, highlighting the increased intensity of the anhydrite peaks (A) (3.50Å), orthorhombic structure, followed by those of gypsum dehydrate (G) (3.06Å), monoclinic structure, majority compounds of the calcium sulfate preparation.

Calcium carbonate, displayed within the upper region of the ground, was feasibly utilized as an extender, polisher and/ or finisher [6–9] and also as a ground layer, being a conceivable memory of the Flemish impact within the employment of chalk ground layers [9,18].

Concerning the technology for the construction of ground layers, this painting contains a thin layer of calcium carbonate of about 14 to 15 μm on the calcium sulfate ground layer.

The ground layer, with about 80 to 140 μm [29], is irregular, and thus, not corresponding to the frequent typology of thicker grains at the base and finer granulometry at the top. In contrast,

a stratigraphic coarse grain is observed at the base, a fine-grained intermediate layer, topped by another coarse-grained top layer.

Given that at the date of the probable making of P5 (between the 1520s and 1530s), as was verified in [30], when some artists began to use calcium sulfate to the detriment of calcium carbonate, this work could have been performed in a phase of experimentation with this material, thus justifying its preparatory irregularity. In P5, the layer of calcium carbonate over calcium sulfate material is so thin that it is not possible to define as being chalk or simply ground calcium carbonate. It may have been applied to regularize the coarse gypsum surface, certainly having the function of leveling the light reflection from the ground layer, given the covering power of calcium carbonate when finely ground and applied in protein binder mixture [31]. Its mixture with carbon black, probably vegetable, considering phosphorus was not identified by the SEM–EDS analysis, darkens this layer, allowing a translucent darker vibration to the overlapping colors. This technique is visible in cross-sectional analysis under carnations and layers of brownish/grayish tonality or in conjunction with the layers composed by azurite, in the skies, and the fabrics. The knowledge that this last pigment could become darker when finely ground, as the Portuguese treatists Francisco de Holanda warned, it is a competing factor for the use of this layer [32]. Thus, azurite could be less milled, keeping all its vibration, but forming a less compact layer. This layer would allow the vibration of the underlying layer, in the case of black tonality, permitting a double reflection of blue and darker colors. The technique of applying a darker layer under the blue is verified in some analyzed paintings, such as the recently studied altarpiece of the Cathedral of Funchal, a collaborative work, specifically in the sample taken from the blue color of the Virgin's mantle in the painting "Descida da Cruz", reveals a probably intentional darker layer with a mixture of lead white and carbon black under the final layer of azurite, as color modellation [28].

Underdrawing features to begin with a geometrical stage made generally by incision and dry charcoal to characterize the position of the figures, a middle stage of form a drawing and a final stage of detail drawing. In IRR and IR photographs it was possible to discover redrawn areas by the creator, as in the case of a few faces but also abandoned drawings, such as some characters found in the sky area or the cross of Christ, left to be painted in a lower position, or the changing of the lion position, previously painted facing the trees, as happens in the illuminated "S.Jerónimo", belonging to the Book of Hours of D. Fernando, made ca.1530–1534, by the Gant–Brugee School (Simon Bening / Horenbout) (Figure 12).

It is also to highlight the fact that in this painting, when performing the IRR analysis, abandoned drawing of a male bust and some faces were found [12]. The experimentation that characterizes this panel is, therefore, noticeable, given that the abandoned drawing, located in the sky area, on the upper left side, does not have correspondence to the painted theme.

Concerning the painting technique, the Master of Lourinhã also delineates in this painting with brownish color the flesh tones as a modeling technique, resembling other coeval paintings and Flemish illuminated manuscripts [18,33–35].

Lead white priming layer is made partially ticker to the sky and landscape areas, and from the results compiled in the report on the analysis of the color layers, the use of the following pigments is revealed: for the red color of the mantle, red and lead white, with ochre added; for the brown colors of the lion, ochre and vermilion, as well as a lower layer with bone black pigment in addition; in the brown of the tree, azurite, lead white, ochre and bone black were detected; for the blue color, azurite with lead white was detected in the robe of the saint and the sky; In the flesh tone of Christ, containing lead white, vermilion and ochre, an underlying layer of dark blue, mainly composed by azurite is found; in the yellow color of the halo, ochre and gold; in the green color of the trees, copper green (verdigris, malachite or copper acetate) and lead white [11].

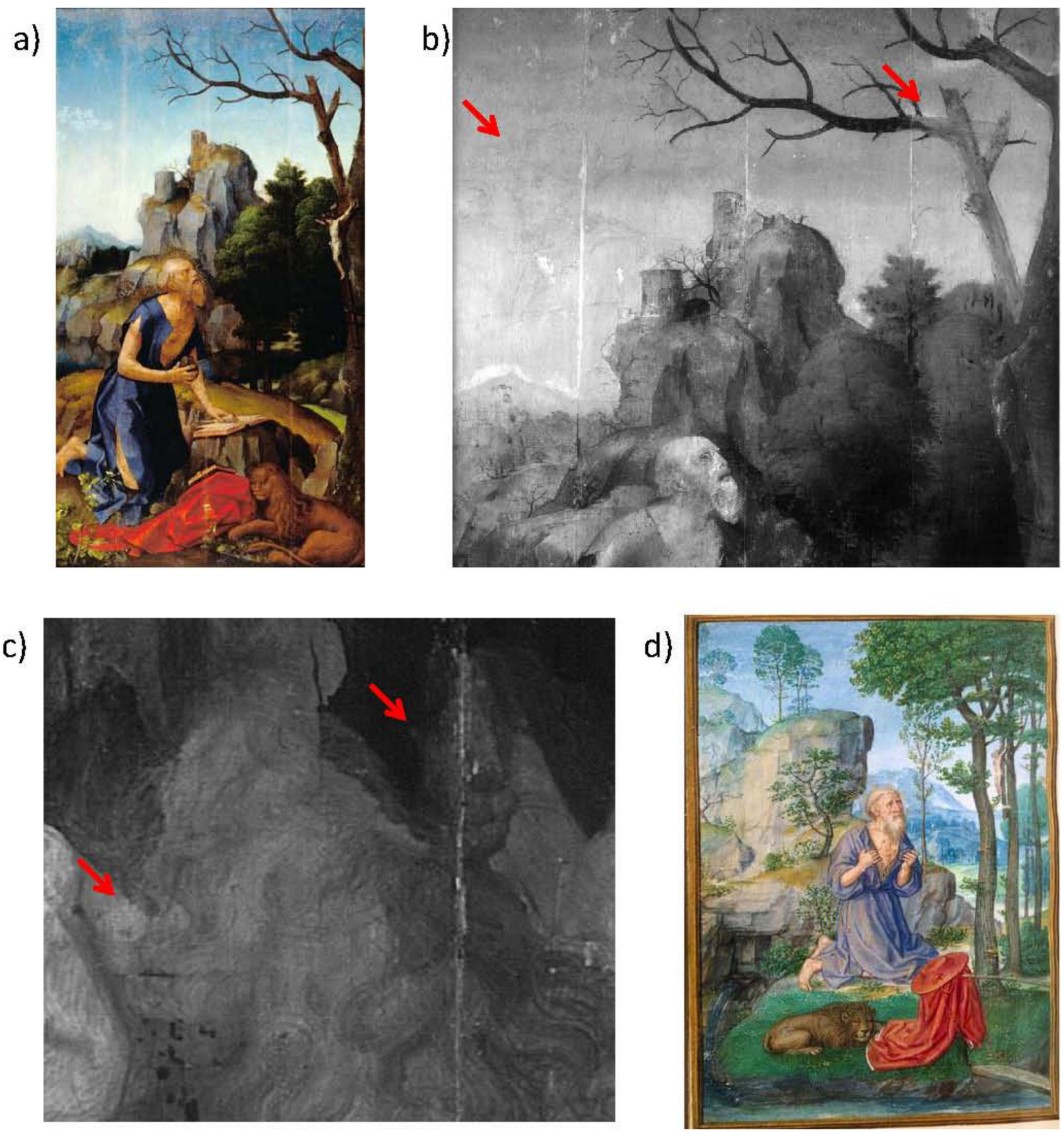

**Figure 12.** (**a**) "S.Jerónimo" (P5); (**b**) IRR characters found in the sky area or the cross of Christ, left to be painted in a lower position; (**c**) changing of the lion position, previously painted facing the trees; (**d**) illuminated "S.Jerónimo", belonging to the Book of Hours of D. Fernando, ca. 1530–1534, by Gant–Brugee School (Simon Bening/Horenbout), Museu Nacional de Arte Antiga, Lisboa, Portugal.

## 4. Conclusions

The Luso-Flemish Master of Lourinhã studied paintings evidence a poor state of preservation and a working technique combining Portuguese and Flemish traditions.

The need to improve conditions of preservation for these paintings evident by the naked eye, were confirmed by this analytical study, namely IRR and IRP area exams and μ-FTIR and μ-Raman analysis, evidencing the presence of metallic carboxylates and plumbonacrite, respectively.

The ground layers of P1–P4 are composed of a double preparatory layer. The first layer is composed by coarse grains, in a matrix of calcium sulfate-anhydrite (*gesso grosso*) with small amounts of gypsum, following the Lisbon workshop technique. The use of minium in this layer (P1–P2) is also to highlight since it was an intentional addition to turn the ground layer tone warmer. The analysis of P5 suggests an experimental use of the ground/priming layers. The irregularity in the application of the double preparatory layer may be justified by the use of calcium sulfate, to the detriment of calcium carbonate, as verified in other artists studied during the transition period of 1530. The thin layer,

probably priming, of calcium carbonate applied over the ground layer of calcium sulfate puts forward the northern European influence tradition while turning the surface for drawing and painting regular.

Similar drawing materials, in particular, the probable use of Iron-based ink suggests a correlation of Master of Lourinhã to the coeval luso-flemish Frei Carlos painting workshop, as defined by art history. However, different ways to sketch the same forms (such as the hands of the figures) and the use of brochantite by Frei Carlos are in evidence Master of Lourinhã as an individual artist. The increase of individual drawing in Master of Lourinhã is remarkable, highlighting an elaborate system of sketching. Collaborative works (Funchal altarpiece) show workshop training by the use of "modeling drawing", less coherent to one draftsman work than the individual paintings of the artist (P1–P5).

The Master of Lourinhã hatching technique in miniaturist style may be correlated to Flemish illumination practices, such as in its aesthetic values, based on Flemish prints and illuminated books. Also, the painting of miniature figures is the closest possible relationship between the artist's painting technique and manuscript illumination.

The painting technique in oil on oak wood for the studied paintings evidence a similar sequence of pictorial construction, from one to three layers, painted from light to shadow (lighter tones firstly painted) using lead white for the light tones and brown ochre and carbon black for the shadow tones, along with the color pigments. Lead white-based priming layers are evident in P1–P4. Gilding technique is performed in these paintings with oil-based mordant and pigments to give volume to the saints' halos. Colouring materials have the palette: lead white, malachite, vermilion, minium, lead-tin yellow, ochre, and azurite. Color modelling layers can be observed in P3–P4 for the carnations and blue layers.

In conclusion, the use of a calcium sulfate matrix in ground layers and the study of the underdrawing and pigment layers unveiled materials and techniques of both Portuguese and Flemish workshops, characterizing the unique work of the Luso-Flemish Master of Lourinhã, a painter formerly in the shadow.

**Author Contributions:** V.A. was responsible for the study proposal, sample collection and preparation, for Raman analysis and for the writing of the text, comparison between artistic and material results; V.S. supervised artistic results; S.V. was responsible for SEM-EDS analysis and results, A.C. (António Candeias) and J.M. supervised SEM-EDS and FTIR results; A.C. (Ana Cardoso) was responsible for FTIR analysis and results; M.M. was responsible for XRF analysis and results; and M.L.C. was responsible for supervising Raman and XRF results and final revision of the text.

**Funding:** This research was funded by research center grant no. UID/FIS/04559/2013 to LIBPhys-UNL, from the FCT/MCTES/PIDDAC and research center grant no. UID/Multi/04449/2013 to Hercules Laboratory. Project ONFINARTS –(PTDC/EAT-HAT/115692/2009) through program QREN-POPH-typology 4.1., co-participated by the Social European Fund (FSE) and MCTES National Fund.PT-FIXLAB and PT-MOLAB from infrastructure E-RIHS (Plataforma Portuguesa da Infraestrutura de Investigação Europeia em Ciências do Património.

**Acknowledgments:** This work was supported by the research center grant UID/FIS/04559/2013, to ARTIS/IHA-FLUL, grant no. UID/FIS/04559/2013 to LIBPhys-UNL, from the FCT/MCTES/PIDDAC and research center grant no. UID/Multi/04449/2013 to Hercules Laboratory. The authors acknowledge SCM Lourinhã for allowing the study of P1-P2 and CM Sesimbra/CESM in the person of Cristina Conceição and to Fr. Eduardo Nobre, Fábrica da Paróquia de Nª Srª da Consolação do Castelo, Sesimbra, Santuário do Cabo Espichel for allowing the study of P3-P4, and to project ONFINARTS –(PTDC/EAT-HAT/115692/2009) through program QREN-POPH-typology 4.1., co-participated by the Social European Fund (FSE) and MCTES National Fund.PT-FIXLAB and PT-MOLAB from infrastructure E-RIHS (Plataforma Portuguesa da Infraestrutura de Investigação Europeia em Ciências do Património. The authors also wish to acknowledge to the HERCULES Lab and José de Figueiredo Lab teams for all scientific collaboration, in particular, Luis Piorro for performing the IRR exams, Maria José Oliveira for XRD analysis and Claudia Pereira at Museums and Conservation library. The authors also wish to thank the National Ancient Art Museum (Lisbon) for granting access to the painting and to Centro de Estudos Históricos da Lourinhã.

**Conflicts of Interest:** The authors declare no conflict of interest.

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
