# Peer review of "A Painter in the Shadow: Unveiling Conservation, Materials and Techniques of the Unknown Luso-Flemish Master of Lourinhã"

_heritage, doi:10.3390/heritage2040169_

Round 1

Reviewer 1 Report

The manuscript refers about the characterization of a pictorial group (Santa Casa da Misericórdia da Lourinhã painting collection), by the use of many different techniques.
This study is interesting, but the text needs major revisions in order to make it clearer (see the details below).

MAJOR COMMENTS
As a first major comment, I did not find any clear description of the motivation in the use of all the techniques employed.
I trust they were all useful to derive important information, but this should be clarified. Otherwise, it seems that the authors used the techniques they have at their disposal.
A clear correlation between materials and characterization analyses is missing, also in relationship with the micro- or macro- sizes, in situ or laboratory use and the techniques spatial resolution. In addition, the use of non destructive and non invasive measurements should be commented, since the study concerns cultural heritage artifacts.
You will find below further questions you should address.

line 83: how many samples were taken? Of which sizes?

line 104: which is the spot size or the spatial resolution for XRF analyses?

line 130: which is the spot size or the spatial resolution for Raman measurements?

line 175: table 1 is not clear. If it is divided by color, it is correct the correlation with different elements detected by XRF, but why putting together all the Raman results?
Moreover, more relevant, references should be added (spectrum ID was used, as written in the text, but references to the pigments association is needed).
Finally, changes in Raman position should be commented. For example, the first band of carbon black is changing from 1322 to 1370 cm-1, and hematite and cerusite peaks are changing too.

line 370: figure 12, and FTIR results should be moved to the results paragraph. Moreover, how was the FTIR band assigment made? No description of FTIR was made in paragraph number 2.

line 372-374: References are needed when assignments are made (like for examples in "displaying Azurite, Kaolinite, Oil and Metallic Carboxylates" or "displaying Azurite, Hydrocerussite, Oil and lead carboxylates").

line 236: this seems the only result obtained by XRD; is this correct? Moreover, the same result could not be derived by Raman spectroscopy?

MINOR COMMENTS
line 52: ";" is needed?
line 56: please, explicit "c."
line 79: please, modify "disbursed"; moreover, all the sentence (lines 79-82) is not clear.
line 87-137: sub-paragraphs 2.2.1 - 2.2.8 are too many; in my opinion it is not useful having so many paragraphs if only few lines are used for the techniques description.
line 88: Optical research microscopy? Which is the difference with the usual optical microscopy?
line 95: please, no comma between photograpy and was.
line 110: what is the meaning of nonheritable for a spectrum?
line 112: "package" is written two times.
line 119: English not clear! "to be anyalayzed"?
line 126: "within the vary 8º to 70º 2θ" is not clear (maybe better: in the 2θ range between 8 and 70 degrees).
line 126: please, no comma after 0.02º.
line 135: "vary" is not clear.
line 157-160: what's the meaning of "reserved" and "reservation" here?
line 251: "photography exams"? Examinations, essays, or just photographs can do.

Final comment: please, check English language for all the manuscript, as well as punctuation, spaces and journal style (see for example: line 65 for spaces, line 69 for bold parts, many figure captions for bold parts, line 157 for style, line 404 for style again).

Author Response

Dear Reviewer

We thank you for your comments, very useful to improve our work.

According to your suggestions, English was improved, along with the introduction (including relevant references),the research design (by adding and clarifying points to the text),methods were improved by revising our English and results were improved by changing table with the results to clarify the findings.

The manuscript refers about the characterization of a pictorial group (Santa Casa da Misericórdia da Lourinhã painting collection), by the use of many different techniques.
This study is interesting, but the text needs major revisions in order to make it clearer (see the details below).

A: Major revisions were performed in the texto (please see text and answers below)

MAJOR COMMENTS
As a first major comment, I did not find any clear description of the motivation in the use of all the techniques employed.
I trust they were all useful to derive important information, but this should be clarified. Otherwise, it seems that the authors used the techniques they have at their disposal.A clear correlation between materials and characterization analyses is missing, also in relationship with the micro- or macro- sizes, in situ or laboratory use and the techniques spatial resolution. In addition, the use of non destructive and non invasive measurements should be commented, since the study concerns cultural heritage artifacts.

A:changed in the text:Considering that this study concerns cultural heritage artifacts, analytical techniques employed were useful to derive important nondestructive and noninvasive measurements information on elementary composition of the materials used by color, specifically adressing µ-XRF analysis. This analysis results were important to define the sampling areas, avoiding overpaintings. Elementary analysiswas complemented by sampling the main colors pallete for each painting and performingSEM-EDS in cross-sections, allowing to understand elementary distribution through layers.This equipment was also useful to recognizethe distribution of the grains, its morphology and granulometry of the ground layer through backscattered electron image (BSE).Ground layer study is essential to determine the material and technical tendencies followed by the artist(ref).Compound analysis on this layer was performed by µ- Raman technique,complemented by µ-XRD when in need to clarify the relative proportion of gypsum/anhydrite facing a calcium carbonate upper layer (P5). In order to understand layering stratigraphy by color, OM was used, and micro-images were taken to compare layering technique by color and by painting. Afterwards, the painting components for each layer were analyzed by µ- Raman technique,in order to recognize the compounds and its state of conservation;these analysis were confirmed by µ-FTIR when hardly identified by µ-Raman technique, as the carbon black or the green pigments, difficulty already noticed by other studies on historical samples (refs).

Area IRP and IRR exams were useful to observe carbon density in drawing by technique and painting, and comparing between paintings, since the same equipment and performance levels were used; these area exams also allowed to distinguish different underdrawing technique for each painting and comparing it between paintings.

You will find below further questions you should address.

line 83: how many samples were taken? Ofwhichsizes?

A:changed in the text:Naked-eye visual examination of the paintings and infrared Reflectography and photography to research on the state of conservation of the painting were performed.elementaryMicro-XRD technique enabled the identification of sampling areas by painting, avoiding overpaintings.One sample of 200-300 µm by color was taken in similar areas for each painting. This allowed to identify components by color and by layer and its state of conservation, also comparing color palettes between paintings. Samples were partly mounted as cross-sections in epoxy chemical compound resin and polished with carbide. One part of each sample was kept to be analyzed by micro-Raman (µ-Raman).

line 104: which is the spot size or the spatial resolution for XRF analyses?

A: a 1.1 mm spot size was used

line 130: which is the spot size or the spatial resolution for Raman measurements?

A:laser spot diameter 1μm

line 175: table 1 is not clear. If it is divided by color, it is correct the correlation with different elements detected by XRF, but why putting together all the Raman results?
Moreover, more relevant, references should be added (spectrum ID was used, as written in the text, but references to the pigments association is needed).
Finally, changes in Raman position should be commented. For example, the first band of carbon black is changing from 1322 to 1370 cm-1, and hematite and cerusite peaks are changing too.

A:Table 1 was improved to be clearer and a table 2 was added to help clarify results.

line 370: figure 12, and FTIR results should be moved to the results paragraph. Moreover, how was the FTIR band assigment made? No description of FTIR was made in paragraph number 2.

A.Corrections were made and FTIR description was added

line 372-374: References are needed when assignments are made (like for examples in "displaying Azurite, Kaolinite, Oil and Metallic Carboxylates" or "displaying Azurite, Hydrocerussite, Oil and lead carboxylates").

Added in table 2 IRUG

line 236: this seems the only result obtained by XRD; is this correct? Moreover, the same result could not be derived by Raman spectroscopy?

A: “Compound analysis on this layer was performed by µ- Raman technique, complemented by µ-XRD when in need to clarify the relative proportion of gypsum/anhydrite facing a calcium carbonate upper layer.” The analyses were performed in different places of ground layer in order to confirm the bigger presence of anhydrite.

MINOR COMMENTS
line 52: ";" is needed?

Changed

line 56: please, explicit "c."

Circa

line 79: please, modify "disbursed"; moreover, all the sentence (lines 79-82) is not clear.

Changed

line 87-137: sub-paragraphs 2.2.1 - 2.2.8 are too many; in my opinion it is not useful having so many paragraphs if only few lines are used for the techniques description.

Changed

line 88: Optical research microscopy? Which is the difference with the usual optical microscopy?

changed
line 95: please, no comma between photograpy and was.

changed

changed
line 110: what is the meaning of nonheritable for a spectrum? 

An error.it was changed in the text

line 112: "package" is written two times.

Changed

line 119: English not clear! "to be anyalayzed"?

changed

line 126: "within the vary 8º to 70º 2θ" is not clear (maybe better: in the 2θ range between 8 and 70 degrees).

changed
line 126: please, no comma after 0.02º.

changed
line 135: "vary" is not clear.

Changed

line 157-160: what's the meaning of "reserved" and "reservation" here? Restricted areas.changes

line 251: "photography exams"? Examinations, essays, or just photographs can do.corrected

Final comment: please, check English language for all the manuscript, as well as punctuation, spaces and journal style (see for example: line 65 for spaces, line 69 for bold parts, many figure captions for bold parts, line 157 for style, line 404 for style again).

English was checked and corrected

Reviewer 2 Report

In this paper, the authors presented the results of a routinely chemical characterization of paint materials and, according to the results achieved, they supported some hypotheses about the influence of two different pictorial traditions on the creation of really valuable paintings by the "Master of Lourinha". 

It is really difficult for the reviewer to fulfil his task since only few significant data are presented in the paper. The results analysis based on IRR pictures (that takes almost 70% of the discussion) is somehow speculative. It is a collection of information that can be read, to my opinion, in many different ways. The clarity of presentation is particularly low, and this makes difficult to appreciate the importance of the conclusions raised by the authors. The integration of further methods of analysis, or the discussion of additional information available from other sources (documental or philological), would have helped to support the author's conclusions. In the present form they look quite weak. 

Micro-XRF and micro-Raman provided useful information about the color palette used by the artist, but it is not clear how this information contributes to clarify the distinctions and similitudes found in previous studies (citing the abstract).

Moreover, the authors recall several times the fact that the paintings are in bad conditions, but throughout the paper almost any evaluation was provided, according to the analysis done, of the active degradation processes. Only figure 10 and 12 provide some information that might have contributed to such discussion, but in the paper this remained unexpressed.  

For these reasons, I believe that the paper in the present form is not suitable for publication on this journal. 

Author Response

In this paper, the authors presented the results of a routinely chemical characterization of paint materials and, according to the results achieved, they supported some hypotheses about the influence of two different pictorial traditions on the creation of really valuable paintings by the "Master of Lourinha". 

Dear Reviewer

We thank you for your comments, very useful to improve our work.

According to your suggestions, English was improved, along with the introduction (including relevant references),the research design (by adding and clarifying points to the text),methods were improved by revising our English, results and conclusion were improved by changing table with the results to clarify the findings.

It is really difficult for the reviewer to fulfil his task since only few significant data are presented in the paper. The results analysis based on IRR pictures (that takes almost 70% of the discussion) is somehow speculative. It is a collection of information that can be read, to my opinion, in many different ways.

A:The result of analysis based on IRR pictures are a conclusion of a large experienceof the team considering the interpretation of Portuguese painting underdrawing.If differences could not be found there was no point on performing these exams.Besides differencing styles between artists and different underdrawing techniques in the same painting, also more or less carbon-based materials can be seen.The aims of the journal focus on knowledge, conservation and management of cultural and natural heritage by sensing technologies. This technique is often used to define similarities and differentiations in underdrawing, being an important exam to deeply understand paintings and painters. Thus, we consider the results on the studied paintings by this technique were valuable to confirm the connection of Master of Lourinhã to Frei Carlos work by the use of the iron-based ink (recognized as fainted drawing in IRR (please see Valadas et al.” New insight on the underdrawing of 16th Flemish-Portuguese easel paintings by combined surface analysis and microanalytical techniques”)),unusual material among portuguese painters at the epoch.Moreover, lacunae localization are easily detected by IRR and IRP analysis, being another way to distinguish between original drawing and restored areas.

The clarity of presentation is particularly low, and this makes difficult to appreciate the importance of the conclusions raised by the authors. 

A:Presentation form was changed and clarity was improved in order to highlight the text concerning the importance of the conclusions. 

The integration of further methods of analysis, or the discussion of additional information available from other sources (documental or philological), would have helped to support the author's conclusions. In the present form they look quite weak. 

A: further methods of analysis are planned, such as lacquer study by HPLS. Although, we need to wait, since further studies on Portuguese paintings are needed in order to compare them. The discussion of additional information available from other sources was added.

Micro-XRF and micro-Raman provided useful information about the color palette used by the artist, but it is not clear how this information contributes to clarify the distinctions and similitudes found in previous studies (citing the abstract).

A: Distinctions and similitudes in previous studies were found when comparing the work of Frei Carlos workshop,the only luso-flemish painter materially studied besides Master of Lourinha (considering the present article).

Moreover, the authors recall several times the fact that the paintings are in bad conditions, but throughout the paper almost any evaluation was provided, according to the analysis done, of the active degradation processes. Only figure 10 and 12 provide some information that might have contributed to such discussion, but in the paper this remained unexpressed.  

A: degradation processes were deepened in the text and brought to discussion.a new subtitle was added “Paintings state of conservation”

For these reasons, I believe that the paper in the present form is not suitable for publication on this journal. 

A:Major changes were made in order to clarify and enrich this research and being suitable for publication on this journal. 

Reviewer 3 Report

Dear Authors,

Thank you for allowing me to read this interesting manuscript on Master of Lourinhã. The abstract clearly defines the research aims, techniques of analysis and different analytical procedures. The description of "Materials and methods" is precise and well-elaborated. Findings are well exposed and the discussion is relevant.

Author Response

Dear Reviewer

We thank you for your comments.English changes were improved.

Round 2

Reviewer 1 Report

The authors have improved the work, adding also non requested parts (like issues on humidity and conservation).

Unfortunately, I think that some results of many techniques (like for example Raman spectra) were not discussed, but used just as a "fingerprint" (to check only the presence of pigments).

For example, my request on Raman assignments was not completely fullfilled.

Table 3 reports the same peaks, without any discussion on the recorded differences in peaks position. Moreover, no description or reference for IRUG database is given.

The text is missing of uniformity in style.

Units are missing in the last but one sentence of paragraph number 2 ("For each spectrum, a spectral resolution of 4 was performed. A working range of 4000-600 and 64 scans were recorded)

Author Response

Dear Refree, we thank you for your revision that has improved our text.

The authors have improved the work, adding also non requested parts (like issues on humidity and conservation).

Unfortunately, I think that some results of many techniques (like for example Raman spectra) were not discussed, but used just as a "fingerprint" (to check only the presence of pigments).

A:Discussion was added to Raman assignments

For example, my request on Raman assignments was not completely fullfilled.

“Moreover, more relevant, references should be added (spectrum ID was used, as written in the text, but references to the pigments association is needed).
Finally, changes in Raman position should be commented. For example, the first band of carbon black is changing from 1322 to 1370 cm-1, and hematite and cerusite peaks are changing too.

A:References added in the text.Changes in raman position were commented

Table 3 reports the same peaks, without any discussion on the recorded differences in peaks position. Moreover, no description or reference for IRUG database is given.

A:IRUG and other database comparing sources were added.

The text is missing of uniformity in style.

A:English was improved in uniformity and style.

Units are missing in the last but one sentence of paragraph number 2 ("For each spectrum, a spectral resolution of 4 was performed. A working range of 4000-600 and 64 scans were recorded)

A:corrected
